# Perception of Daily Time: Insights from the Fruit Flies

**DOI:** 10.3390/insects13010003

**Published:** 2021-12-21

**Authors:** Joydeep De, Abhishek Chatterjee

**Affiliations:** 1Department of Pharmacology, University of California San Diego, La Jolla, CA 92093, USA; jde@ucsd.edu; 2Institute of Ecology and Environmental Sciences of Paris, iEES-Paris, INRAE, Sorbonne Université, CNRS, IRD, UPEC, Université de Paris, 78026 Versailles, France

**Keywords:** drosophila, circadian, clock neurons, locomotor activity rhythms, entrainment, sleep-wake cycles, clock output

## Abstract

**Simple Summary:**

The ability to perform time-dependent functions has been linked to survival in animals. For a prey, to forage when many predators are active, can be deadly. Performing the right task at the right time is dependent on the animal’s ability to track daily time. Perception of time in the animal kingdom dates long back on the evolutionary time-scale. From the unicellular prokaryotes to humans; we all tune our physiology and behaviours according to the time of the day. To do this in the face of unreliable environmental parameters, we use an internal timekeeper. The state of the timekeeper is readjusted daily, by feeding weighted updates about the external factors that oscillate in nature, e.g., light and temperature. In this review, using the example of the fruit flies, we discuss how animals perceive the external and internal times and connect them in the brain.

**Abstract:**

We create mental maps of the space that surrounds us; our brains also compute time—in particular, the time of day. Visual, thermal, social, and other cues tune the clock-like timekeeper. Consequently, the internal clock synchronizes with the external day-night cycles. In fact, daylength itself varies, causing the change of seasons and forcing our brain clock to accommodate layers of plasticity. However, the core of the clock, i.e., its molecular underpinnings, are highly resistant to perturbations, while the way animals adapt to the daily and annual time shows tremendous biological diversity. How can this be achieved? In this review, we will focus on 75 pairs of clock neurons in the Drosophila brain to understand how a small neural network perceives and responds to the time of the day, and the time of the year.

## 1. Introduction

Organisms keep time to survive. The time of the day can be perceived by observing the changes in the environment or rather anticipating the changes in the environment. Circadian clocks have evolved to enable organisms to perceive time in anticipation of environmental changes. In general, daily variations in light (spectral composition, intensity, polarization) and temperature are the environmental cues that most animals use to perceive the time of the day. The inputs with such environmental signals reach the brain (or, peripheral) clocks where they are interpreted and integrated, and the physiology or behaviours are accordingly timed by the clock.

It is interesting to ponder upon the question of whether it would have been of greater advantage if organisms had a system that could do the job of time-keeping by just passively responding to environmental changes. One way to interpret the evolution of internal self-sustained time-keeping systems would be that the changes in the environmental conditions might not be reliable and sufficient. This is true for animals inhabiting the arctic region where temperature and light conditions do not have much daily variations. It is also true for a midsummer’s day marked with an afternoon storm that temporarily reduces light availability and dips temperature. Another reason for the evolution of self-sustained oscillators is the ability to predict environmental changes. It might not always be productive if the animal has only the ability to respond to changes in the environment; especially when there is a mortal consequence of not being able to predict it. For example, in terms of a strategy for survival, it is intuitive that a prey needs to be able to predict the foraging timing of its potential predator(s) rather than just possessing the ability to take action while it has already encountered the predator. In fact, maintenance of an internal temporal order, i.e., segregating interfering biochemical processes and clubbing together interdependent biochemical processes is thought to be the main driving factor behind the evolution of the circadian clock.

The near-ubiquitous occurrence of daily rhythmic behaviours across taxa being largely controlled by circadian clocks provokes an enormous academic need to understand the genetic, molecular, neural, and evolutionary mechanisms of such phenomena to decipher how perception of time is encoded in animals. Drosophila or the fruit flies have provided many insights on how organisms adjust their physiology and behaviours according to the time of the day.

Drosophila exhibits several signatures of its perception of time. Flies perform many behaviours that are periodic, and hence, necessarily, time-specific. Flies are crepuscular: they are active only during certain parts of the day, unlike diurnal animals which are active throughout the day, and nocturnal animals, which are active throughout the night. Drosophila is active around the dawn and the dusk transitions of the day; giving rise to two distinct activity peaks in the morning and evening [1,2,3]. In parallel to locomotion, flies also show sleep-like states and exhibit daily sleep patterns with most of the sleep occurring during the night with an additional bout of sleep in the middle of the day (siesta) [4]. Flies, as a population, do also exhibit daily bursts of emergence of adults from the pupal cases early in the morning [5,6,7]. A peripheral clock in the prothoracic gland, with the help of the central clock in the brain, controls the rhythm in adult emergence [8]. There are daily variations in taste [9], feeding [10], in female mating [11] and male courtship [12,13,14]. Flies have diurnally varying olfactory responses [15]. This is known to be regulated by the clocks in their antenna which are both necessary and sufficient in generating such rhythms [16]. A lesser investigated rhythm in flies lies in the daily egg-laying behaviour of Drosophila females which do not rely on the canonical clock centres in the fly brain; namely, the Pigment Dispersing Factor (PDF) expressing lateral neurons in the brain [17,18]. Maximum bouts of egg-laying happen in the later afternoon [17]. Another brain-independent circadian clock residing in the epidermal cells regulates a daily rhythm in the cuticle deposition [19]. In addition to these time-dependent behaviours, the photoreceptive and autonomous circadian clocks present throughout the fly body including proboscis, antennae, legs, and wings are suggestive of many more daily timed phenomena in the fly [20] (Figure 1).

## 2. Drosophila Circadian Timers

Drosophila has around 150 neurons in the brain which are defined as ‘clock neurons’, as each of them possesses a molecular oscillator. The molecular clock is centred around the CLOCK/CYCLE (CLK/CYC) activator complex that binds to E-box elements in the promoters of period (per) and timeless (tim) and upregulates their transcription. These transcripts, upon translation, form a heterodimer and migrate to the nucleus with the help of kinases such as Shaggy (SGG) and Casein Kinase II (CK2). Association of the PER-TIM complex with CLK-CYC leads to DOUBLETIME (DBT)–dependent hyperphosphorylation of CLK and eventual cessation of CLK/CYC’s transcriptional activity. Consequently, per and tim stop to be transcribed. CK2, DBT, and NEMO kinases, with the help of ubiquitin ligases degrade the repressor complex, ensuing a new beginning of the feedback loop. Multiple checkpoints and inbuilt delays ensure a 24 h rhythmicity of the clock genes and proteins with the help of post-transcriptional and post-translational modifications [21].

The circadian clock neurons of the brain interact with each other as a distinctly coupled clock network. Irrespective of having the same molecular oscillations, clock neurons have different functions, coloured by the interactions among different oscillators. Moreover, subtle but clear differences in the molecular make-up of their clockwork have been noted [22]. Drosophila clock neurons can be divided into lateral and dorsal neurons. Among the lateral neurons, the small and large ventro-lateral neurons (s-LNvs and l-LNvs) express a neuropeptide called Pigment Dispersing Factor (PDF) [23,24,25]. Other lateral neurons (LNds and LPNs) and dorsal neurons (DNs) do not express PDF but several of them contain the receptor for PDF (PDFR) [26]. PDF acts as one of the main modes of communication among the clock neurons and is released from the s-LNvs in a periodic fashion [25]. PDF from the LNv neurons communicate with the LNds and DN1s via PDFR and regulate their functions in light-dependent manner [27].

The dual oscillator model, developed by Colin Pittendrigh and Serge Daan in 1976, suggests that there are two different oscillators on which morning and evening activities could rely [28]. This was originally proposed as a consequence of an observation that, in nocturnal rodents, the activity rhythm splits into two components with different periods under certain conditions. Pittendrigh and Daan proposed that the two components come from two different oscillators (hence, dual oscillators). They also floated the idea that these two distinct but coupled oscillators could enable adaptation to different photoperiods via morning clock entraining to lights-ON transition and evening clock entraining to lights-OFF transition. Dual oscillators have not yet been unequivocally localized in the nocturnal rodent’s Supra-Chiasmatic Nucleus (SCN) [29]. Nevertheless, when the idea of a dual oscillator model was tested in Drosophila in 2004, two laboratories independently reached the conclusion that distinct oscillators in the brain drive morning and evening activity bouts [1,2]. Grima and others rescued functional clock (by the expression of period gene in a period-null mutant fly) in specific subsets of the clock neurons and showed that while PERIOD is rescued only in the PDF+ neurons (henceforth, M neurons), the morning anticipation is restored whereas PERIOD in both PDF+ and PDF− neurons rescues evening activity. Stoleru and others, on the other hand, genetically ablated certain subsets of the clock neurons and showed that there are two oscillators for two activities. These and a few consecutive studies, overall, indicated that PDF+ neurons (especially, s-LNvs) working in series with certain DN1ps drive morning activity, whereas PDF− LNds and few DN1s working in parallel drive evening behaviour [27,30].

## 3. The Time Cues

The perception of time depends crucially on the ability to track natural time cues. Nonetheless, being able to track the changes in the environment is necessary, but not enough. The organisms then need to synchronize their physiology and behaviours to the most ‘favourable’ time. Indeed, the major goal of biological oscillators is to synchronize with the periodically changing environment. There could be a vast array of stimuli to which an organism might need to synchronize its clocks to, but the most widely studied stimulus is light. The endogenous period of the biological systems or τ (tau) has to be daily adjusted to the periodicity of the time cue (T) or the zeitgeber (zeit = time; geber = giver). In case of a 12 h light–12 h dark cycle (LD 12:12), the T is 24 h and the near-24 h period of the circadian system has to be adjusted to the 24-h periodicity of the zeitgeber, every day. One way in which entrainment occurs is known as phasic or non-parametric entrainment since the organism’s entrainment depends on certain phases (e.g., dawn and dusk transitions in case of light as a time cue). This phasic entrainment can be aptly illustrated by Phase Response Curves [31] or PRCs, which are graphical representations of phase-changes in response to a pulse of light at a given phase. To generate a PRC, light pulses are applied at specific circadian phases in otherwise constant darkness (or, dark–dark; DD). The response of the clock to these pulses is manifested in terms of changes that the measurable output achieves [28,32]. In Drosophila, during the early subjective night, upon light pulses, the output phases (e.g., the onset of the locomotor activity) delay, and during late subjective night, the output phases advance. There are almost no phase changes during the subjective day. There could also be non-photic PRCs; such as phase-responses to temperature, locomotion, and social stimuli [28,33,34]. PRCs might differ from species to species, and from low-amplitude (type 1; such as in wild-type Drosophila in response to light pulses) to discontinuous and high-amplitude (type 0; such as in mutant pers Drosophila in response to light pulses) [35].

Although huge explanatory power has been attributed to phasic or non-parametric PRCs in explaining entrainment to photic pulses, it is not the only way one might perceive how entrainment occurs. One crucial assumption in the non-parametric mechanism is that the animals are supposed to daily experience, at least one of the two light–dark transitions (dawn and/or dusk). However, there are animals such as ground squirrels in Europe (*Spermophilus citellus*) that experience neither of these two phases in the wild [36]. These animals are diurnal, burrow-dwelling and they hardly come out of their burrows around the daily light–dark transitions [36]. These animals probably depend on daily changes in the light intensity during the course of the day, therefore ruling out the possibility of being entrained by the crucial transition phases [37]. Such an alternative mechanism dictated by the tonic effects of light that supposedly change the velocity (i.e., parameter) of the internal oscillator is known as parametric entrainment.

### 3.1. Light Inputs

Phase resetting of the Drosophila brain circadian clock by light occurs through the intracellular photoreceptor CRYPTOCHROME (CRY) which is expressed by a number of clock neurons [38,39]. CRY is a UV-blue photoreceptor (420 nm) and it can function both cell-autonomously and non-cell-autonomously [38,39]. CRY transmits light information to the clock through its light-dependent interaction with TIMELESS (TIM) that engages the F-box protein JETLAG and triggers TIM degradation [40,41,42]. Constant light acting via CRY induces behavioural arrhythmicity and disruption of the molecular clock [43]. Light pulse in early subjective night delays the phase of the rhythm and the same in early subjective day advances it. This phase shift depends on TIM protein levels and its location in the cell [44]. Individual CRY(+) oscillators in different clock neuronal groups also show distinct kinetic signatures of light response and phase retuning following a transient asynchrony [45].

The clock is fed with multiple photic information so that it can update its endogenously computed time of the day. In other words, Drosophila uses parallel and partly redundant mechanisms to receive light information. CRY is not the only photoreceptor that sends light signals to the clock, since cry01 flies can also ‘phase-shift’ to light pulses [46] albeit with much less efficiency. Other than CRY, the only known clock-cell resident photopigment is Rh7, a UV-violet sensitive rhodopsin molecule found in few DNs [47,48]. Notably, Rh7′s role in circadian entrainment is debatable; because the brain Rh7, by itself, fails to entrain the clock [49]. Flies can also synchronize with the external day–night cycles using photic inputs from the compound eyes, ocelli, and H-B eyelets (Figure 2). The cholinergic output of these photoreceptor neurons is not sufficient for circadian re-entrainment, but their histaminergic output is. Rh6-expressing photoreceptors pool visual input and communicate with the clock neurons [50]. Future research should decipher whether specific Rh-expressing photoreceptors connect with specific clock neurons, whether their connection is direct and time-varying, and what is the ultimate molecular clock target of the visual light inputs.

### 3.2. Temperature Inputs

Temperature could also play the role of a crucial time cue for the circadian clocks. Temperature cycles, even as low as 2 °C, could rescue behavioural rhythmicity induced by constant light [51]. Many of these temperature cycle studies in flies were conducted in constant light (LL) condition, where constantly available light degrades TIM through CRY permanently without leaving any possibility of PERIOD (PER) and TIM to rebuild their levels in order to be cyclic. In the backdrop of this apparent absence of molecular clock in LL, temperature cycles rescue molecular oscillations of PER and TIM [51]. Temperature-dependent alternative splicing of tim has been proposed to set the levels of TIM and contribute to clockwork maintenance under temperature cycles in LL [52]. Although both light and temperature cues impinge on the clock to entrain it, neuronal loci of their function among the clock neurons probably do not overlap (Figure 2). Dorsal (DNs) and lateral posterior neurons (LPNs) are crucial for temperature entrainment [51,53]. As discussed before, entrainment to light happens mainly in the lateral neurons [2]. Light and temperature, therefore, probably work on two different sets of clock neurons and the efficient coupling between them enables Drosophila to keep proper phase relationships to light and temperature cycles [53] as found in nature. A subset of the DN1s have been recently described as cold-activated neurons and they receive thermal information from two peripheral organs—the chordotonal organs and the arista [54].

**Figure 2 insects-13-00003-f002:**
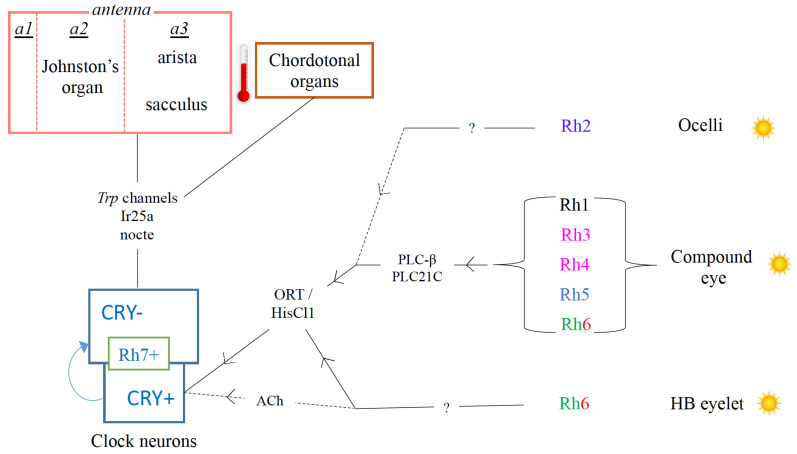
Inputs to the clock. The two major time-cues for the insect circadian clocks are light and temperature. Light is received by the visual system and by intracellular CRY and Rh7 present in some of the clock neurons in the brain. The light inputs from the visual system is first transduced by different rhodopsins (Rhs) in the compound eye, ocelli and HB eyelet, and finally reaches the clock through parallel, multi-synaptic circuits involving histaminergic, cholinergic (and potentially peptidergic) pathways. Temperature receiving sensory units present in the body (Chordotonal Organs), or the tripartite (a1–a3) antenna of the head (particularly, in the Johnston’s organ, arista and sacculus) feed information to mostly the CRY(−) clock neurons via indirect connections. Thermal Trp channels (TrpA1, pyrexia), ionotropic receptors (Ir25a) and genes that control the structural integrity of the chordotonal organs (nocte) have been implicated in temperature entrainment (reviewed in [55]).

### 3.3. Adaptation to ‘Natural’ Input Conditions

Light and temperature, as discussed above, are potent time-cues for the circadian clock. Since the animals in the wild experience different daily varying time-cues simultaneously, it provided enough reasons to study the impact of simultaneously changing light and temperature inputs. These two cues, when put together in phase, had a synergistic effect on the entrained activity-rest behaviour. Flies exhibited sharper M and E anticipations and the molecular clock components had a higher amplitude cycling [56]. On the other hand, flies follow the light cue when light and temperature are put out of phase with respect to each other, thereby inducing a conflict [57,58]. This suggested that light is a stronger time-cue than temperature. Interestingly, this preference for light was subdued in flies that lacked CRY. Light input components like CRY seem to suppress entrainment to temperature [59]. In addition to the simultaneous occurrence of varying light and temperature with a staggered phase, the nature of these daily varying cues also crucially differs in the wild. Perception of time, therefore, might be very different between the laboratory and the outside. Nonetheless, flies continue exhibiting rhythmic rest–activity and eclosion behaviours even under complex semi-natural conditions outside the lab, albeit with significant changes in their behavioural patterns [60,61,62,63,64,65]. Although PER and TIM oscillate in the clock neurons in the flies outside the lab, the coordinated nuclear entry of PER and TIM seen in most laboratory light/dark cycles was not seen under summer conditions [60]. We are yet to know, faced with a long phase-lag between PER and TIM oscillations, how the molecular clockwork continues to generate rhythmic output under semi-natural conditions.

Drosophila melanogaster, evolutionarily speaking, comes from the tropics. However, today, it is found in higher latitudes too. Studying the adaptation strategies that Drosophilid flies used to colonize northern Europe, for example, is important to understand how animals perceive time under different environmental conditions. Unlike the tropics where the day-length does not vary much across the year, the northern flies experience long summer days. These northern species of Drosophila under summer-like conditions show a broader afternoon/evening activity band without a sharp peak, unlike melanogaster flies in the equinox [66,67,68]. In contrast to having a stronger clock to deal with drastically long day-lengths [69,70], many of these species have clocks with weakened light sensitivity [67]. A number of the northern flies do not express PDF in the s-LNvs and CRY in the l-LNvs. Nevertheless, their atypical neurochemistry is not tightly correlated with their ability to adapt to the long and mild summer days typical of the subarctic zone [71].

In Drosophila, the adaptation of the circadian clock to different environments has been historically studied mostly in terms of changes in the photoperiod as long day represented summer and short day represented winter. Indeed, flies do entrain to changing photoperiods by adjusting their activity phases. In long day, for example, the M peak advances and the E peak delays compared to equinox day. In accordance with this, light was proposed to accelerate the M oscillator and decelerate the E oscillator, although experimental results hinted at a more nuanced scenario [30]. Light not only just impacts on the molecular clock itself but also seasons their outputs. Light activates E output whereas it inhibits M output [72]. As a correlate of the day-length, M and E clocks also switch controls between themselves as to who governs the variably-coupled clock system globally [3]. This work revived the idea that circadian clocks do also keep time of the season. That the M clock enslaves E oscillator in short days has been unequivocally demonstrated by later studies [27], but their proposition that the E clock on long days enslaves the M oscillator still awaits empirical validation. Furthermore, day-length is not the only signature of seasonal changes on the earth. Daily average temperature also changes across seasons, especially in the temperate climates where flies are plentifully abundant today. Drosophila has evolved and consequently spread over vast geographies and climates which must have necessitated the evolution of seasonal timing system based on many available and trackable seasonal cues other than just the length of the day.

Phenomenal work done in the laboratory of Isaac Edery, over the past few decades, has been able to shed some light on how changes in temperature could be adapted by the fly through its circadian clock. They showed that temperature-dependent splicing of a clock mRNA is crucial for thermal adaptation. Splicing efficiency of dmpi8, the 3′ intron in per mRNA, is subject to temperature changes [73]. In low temperatures, the dmpi8-spliced per mRNA is more than the non-spliced ones. Shortening of photoperiod, like cooler temperature, enhances dmpi8 splicing [74,75]. Through an unknown mechanism, efficient splicing leads to a rapid daily increase in the per transcripts and earlier evening activity. In fact, splicing itself seems to be important rather than the retention or removal of the dmpi8 intron. Moreover, dmpi8 of per forms a functionally integrated genetic unit with the partially overlapping but reverse-oriented gene dyw (dayawake) which functions as a day-specific anti-siesta gene. Remarkably, dyw expression is stimulated in trans via cold-enhanced splicing of the dmpi8 intron [76]. As daily temperatures become cooler and the chance of daytime heat exposure reduces, dmpi8-dwy acclimator unit acts to prepone evening activity. This, intuitively, finds some ecological basis since it is not advantageous for a fly to be very active in the hot midday as it risks desiccation. In reality, however, very high midday temperature ≥ 35 °C promotes locomotor activity presumably for escape from the hot microhabitat [65], which ultimately highlights temperature range-specific adaptation strategies that the vinegar fly employs. It was thought that temperature regulates per transcripts, whereas light, in low but not high temperature, augments tim expression [77]. Recently, three groups demonstrated that alternative splicing of tim transcripts provides a second temperature-sensing mechanism for the clock. High temperature upregulates tim-*medium* (tim-*M*) level, and lower temperatures favour tim-*short and cold* (tim-*sc*) and tim-*cold* splice-forms. Increased retention of an intron in tim-*M* during the hotter daytime serves to decrease TIM levels and delay the accumulation of TIM. Reciprocally, tim-*sc* gives rise to a markedly shorter form of the TIM protein that advances the clock through poorly characterized mechanisms [52,78,79]. Isoform switching of tim is perhaps most relevant for entrainment as well as adaptation to high-amplitude temperature cycles, a characteristic of the summer days in a sub-tropical climate.

## 4. The Output of the Clock

Perception of time is necessary for physiological actions to be aptly timed. The circadian clock that regulates activity–rest rhythms resides in the brain of a fly. From the clock proteins inside the clock neurons embedded within a network in the brain, how this message is translated to the cyclical changes in organismal physiology is a big question that remains largely unanswered. There have been, nevertheless, few attempts in order to understand the neural and molecular pathways toward the output(s) of the clock. The circadian molecular rhythmicity (core clock) provokes rhythmic changes in output molecules’ abundance, localization, release, or activity. The circadian pacemaker neurons (namely, the s-LNvs), which run endogenous circadian rhythms, are shown to have daily oscillations of their membrane potentials [80,81]. This has also been seen in another subset of clock neurons: the DN1s [82]. These two subsets exhibit spontaneous spiking with a daily variation; more in the morning (high rate of firing) and less in the first half of the night (low rate of firing). Their pattern (temporal code) of firing also exhibits circadian variation. Sodium and potassium ion conductances vary in a daily fashion too. The sodium conductances vary in anti-phase to potassium conductances [82]. The sodium channel involved is called Narrow Abdomen (NA) [83] which is characterized as a sodium leak channel (NALCN). NA is rhythmically delivered to the axon membrane via clock-regulated NLF-1 (NCA-localisation factor), providing direct evidence that the functional clock talks to the membrane [82]. Rhythmic membrane excitability of the LNv drives daily rhythm in PDF release from the axons. Unexpectedly, instead of an electrochemical, a biochemical (inositol triphosphate, IP3-dependent) mechanism times PDF release from the LNv soma, which precedes axonal release by hours [84]. However, the behavioural significance of the fact that the pacemaker neuron releases neuropeptides from different sites at different times of the day remains elusive.

Similar to the oscillations in the membrane conductances, the synaptic plasticity of the pacemaker neurons is also subject to diurnal variations. s-LNv projections show maximal projection spreads around dawn which also coincides with its depolarized membrane with higher firing rate as discussed above. GTPase Rho1 is the molecular factor identified to be controlling this rhythmic phenomenon which, in turn, is regulated circadianly by clock-regulated transcription of Rho1 GEF (Purathrophin-1-like) [85,86,87,88]. Whether the structural plasticity of the M and E neurons’ axonal arbor dictates locomotor output rhythms or encodes incoming temperature input remains unresolved [89,90].

The unimodal mRNA and protein oscillations in the clock neurons are causally associated with two daily activity peaks. Also, other than a few minor differences between the molecular clocks among the M and E neurons, they are mechanistically very similar. The phases of PER and TIM oscillations, largely, do not alter depending on the nature of the clock neuron. A long-standing question that has intrigued circadian fly biologists, therefore, is how the same clock in some neurons gives rise to a behaviour in the morning and in some other neurons, in the evening. Pioneering studies from Paul Taghert’s group have shown that there are calcium oscillations in the clock neurons which are phase-separated between the morning and evening neurons, such as, the calcium peak in the morning neurons occurs before morning and the same in the evening neurons occurs before evening. This provides a bridge between the molecular oscillations and the behaviour, in that, the functional segregation between the M and E neurons occurs in those neurons themselves, downstream of the clock [91]. Multi-synaptic pathways independently link the M and E oscillators as well as the DN1 clock neurons to the ring neurons of the ellipsoid body [92,93,94,95]. However, how the ring neurons integrate clock inputs and sculpt the two daily bouts of activity are far from being clear.

There are multiple downstream targets of the clock neurons that have been identified. Justin Blau’s group identified a pair of lateral horn neurons expressing the neuropeptide leucokinin (LK) being crucial for the propagation of clock information pertaining to DD rhythms—the LNv neurons drive rhythmic changes in LK neurons’ activity, and the LK neurons, in turn, communicate with the LK-receptor expressing neurons in the central complex through inhibitory connections [96]. Null mutations in a K channel, slowpoke (slo) [97], or slowpoke binding protein (slob) [98] produce behavioural arrhythmicity under constant darkness while the central clock remains uninterrupted, intuitively promoting their candidature for potential clock output genes. Further studies showed that the behavioural arrhythmicity in the slo mutants comes from abnormal PDF projections near the dorsal clusters (DNs) of the clock neurons [99]. The cycling of SLOB, especially in the Pars Intercerebralis (PI), was shown to be dependent on Clk^jrk^. Similarly, rescuing slo selectively in the PI restores the behavioural rhythmicity [100]. In accordance with the PI-specific roles of the clock output molecule slo, ablation of PI neurons also induces arrhythmicity [101], highlighting that PI is one of the major relay centres for clock information. DN1s and a subset of the LNds were shown to contact PI neurons directly, and their connection strength showed daily oscillations [101,102]. DH44 (diuretic hormone 44), expressed by specific PI neurons, was shown to be a circadian signalling molecule, and DH44+ PI neurons receive inhibitory inputs from the clock neurons [102]. Downstream of these DH44+ PI neurons, hugin-neurons of the suboesophageal zone, in turn, talk to the ventral nerve cord and propagate clock information [103]. Notably, the linear peptide relay orchestrated by DH44+ PI neurons is modulatory in nature and may not constitute the sole output pathway for locomotor rhythms in DD. The role in rest-activity rhythms of other PI peptides-insulin-like peptides (ILPs) and SIFamides—are not yet clear, they might be more relevant for feeding rhythms [102,104]. Every known output neuropeptide (LK, DH44, ILPs, SIFa) also affects sleep amount or quality, hence their circadian influence on locomotor activity rhythms has been often hard to disentangle. Surprisingly, none of these neuropeptides are indispensable for the evening activity under LD cycles.

## 5. Conclusions

Animals perceive time by measuring durations of or between events, by noting the sequence of events, and also by responding to changes that are periodic. This review summarizes how the model insect, fruit fly, understands time using the third approach. This approach achieves proactivity, i.e., overcomes the limitation of being purely reactive, by generating an internal reconstruction of the cyclic external environment. Internal representations are formed by the CLK-CYC-driven oscillating gene expressions which can circularly run on their own for several days. On encountering a new day–night cycle, the phase (and period) of this biological clock gets redefined by the agents of the environment which change periodically. For changes with different periodicities in the environment, animals have evolved clocks with different periodicities, too: starting from the clocks that help migrating birds spanning seasons, to clocks that command pulsatile hormone release (τ~minutes) from the neurosecretory cells of the insect brain. The ways through which the circadian clock senses the outside world offer a general model for understanding the entrainment mechanisms of infradian and ultradian clocks, not least because parts of the molecular clock circuitry are co-used by non-circadian oscillators.

## Figures and Tables

**Figure 1 insects-13-00003-f001:**
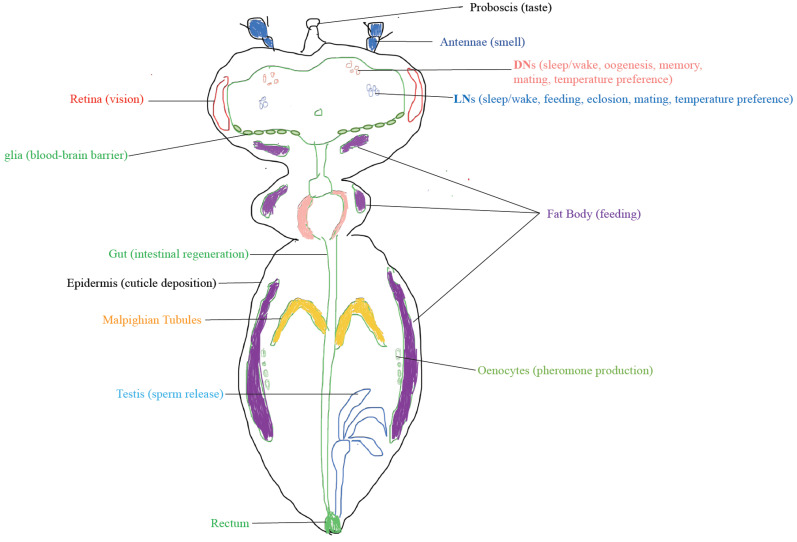
Pervasive clock-driven physiologies. Clock-regulated diurnally varying biological processes are observed all over the fly body. While the clocks in the brain (LNs and DNs) regulate multiple behaviours such as sleep/wake, mating and feeding, the peripheral clocks often regulate local physiologies—either in consultation with the central brain clock, or autonomously. However, for a number of peripheral clocks, the physiological outputs that they regulate, are yet to be known.

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
