# Peer review of "Perception of Daily Time: Insights from the Fruit Flies"

_insects, 2021, doi:10.3390/insects13010003_

Round 1

Reviewer 1 Report

The author reviews the ample literature on circadian timing in the fruit fly Drosophila melanogaster supplying a good overview of the most important findings. It would have been very nice for the general reader to also include a Figure about the most important clock output pathways in the midbrain including respective neuropeptides/neurotransmitters relevant for timing of rest-activity cycles and feeding.

Specific comments:

L57 Peripheral clocks…control

L59 …variations in taste…

L61 Flies´ olfactory responses vary diurnally (Krishnan et al., 1999) and are regulated by the clocks in their antenna…

Fig. 1 legend:…Clock driven physiologies and organs {alternatively: mention physiologies for all organs listed}

L85 …orchestrated via kinases… {please specify}

L86 “Their” reference is not clear, please specify

L89 …degrade…

L94 formulate more clearly such as “The circadian clock neurons of the brain interact with each other as distinctly coupled clock network. “{they form a network rather than having a network sandwiched between the clock neurons}

L102 Is it really true that only some of them and not most of circadian clock neurons express the PDFR?

L105 please add reference for periodic release of PDF

L107 …espoused? Please replace by ”first suggested” or “developed”….

L116 ….suprachiasmatic nucleus…

L122 rescued

L142 …since the organism´s entrainment depends on…

L146…at specific circadian phases of the stimulus under constant conditions…

L150 please clarify, e.g.…the onset of the locomotor activity rhythm is delayed, while light pulses during the late subjective night cause advances.

I would also cite the terminology of Johnson1992 for the explanation of PRCs

L171 …really primarily? Isn´t there a difference depending on peripheral and central clocks, and also on intensity and frequency of the photic stimulus?

L188…sends light signals

L191 …role in circadian…

L192…please replace “certainly” with “because”

L202…2 degree Celsius?

L215…two different targets of clock neurons…

L217…relationships to…

L226 …(chordotonal organs),…

L234…cycled at higher amplitudes…

L239 ...seem to…

L275 …whereas it inhibits…

L309-317 Please name first the abbreviations of the different splice forms of tim transcripts before explaining their respective functions.

L320 …activity-rest rhythms…

L332-338 please add all relevant citations

L340 …(Klose et al., 2021).

L345 …around dawn…

L359-363 Please discuss the results/discrepancies of/between findings of the Taghert lab that measured Ca++ levels in the somata and the finding by Klose et al., 2021.

L382…talk to

L386 ..insulin-like peptides (ILPs) and SIFamides…

L378 …Please discuss also the findings by Ruiz et al. 2021.

Author Response

Specific comments:

L57 Peripheral clocks…control

It is corrected.

L59 …variations in taste…

It is corrected.

L61 Flies´ olfactory responses vary diurnally (Krishnan et al., 1999) and are regulated by the clocks in their antenna…

It is corrected.

Fig. 1 legend: Clock driven physiologies and organs {alternatively: mention physiologies for all organs listed}

It is corrected.

L85 …orchestrated via kinases… {please specify}

Done.

L86 “Their” reference is not clear, please specify

Done.

L89 …degrade…

It is corrected.

L94 formulate more clearly such as “The circadian clock neurons of the brain interact with each other as distinctly coupled clock network. “{they form a network rather than having a network sandwiched between the clock neurons}

It is corrected, accordingly.

L102 Is it really true that only some of them and not most of circadian clock neurons express the PDFR?

The modified sentence reads: Other lateral neurons (LNds and LPNs) and dorsal neurons (DNs) do not express PDF but several of them contain the receptor for PDF (PDFR) (Im & Taghert, 2010).” Indeed, PDFR is not expressed by every clock neuron.

L105 please add reference for periodic release of PDF

It is added.

L107 …espoused? Please replace by ”first suggested” or “developed”….

It is modified accordingly.

L116 ….suprachiasmatic nucleus…

It is corrected.

L122 rescued

It is corrected.

L142 …since the organism´s entrainment depends on…

It is modified.

L146…at specific circadian phases of the stimulus under constant conditions…

The modified sentence reads: To generate a PRC, light pulses are applied at specific circadian phases in an otherwise constant darkness (or, dark-dark; DD).

L150 please clarify, e.g.…the onset of the locomotor activity rhythm is delayed, while light pulses during the late subjective night cause advances.

Modified accordingly.

I would also cite the terminology of Johnson1992 for the explanation of PRCs

Cited.

L171 …really primarily? Isn´t there a difference depending on peripheral and central clocks, and also on intensity and frequency of the photic stimulus?

We have added the word 'brain' and removed the word 'primarily'. We assess the non-CRY phase resetting in the following paragraph. The sentence under scrutiny reads now as: "Phase resetting of the Drosophila brain circadian clock by light occurs through the intracellular photoreceptor CRYPTOCHROME (CRY) which is expressed by a number of clock neurons"

L188…sends light signals

Corrected

L191 …role in circadian…

Corrected

L192…please replace “certainly” with “because”

Corrected

L202…2 degree Celsius?

Corrected

L215…two different targets of clock neurons…

The modified sentence reads: Light and temperature, therefore, probably works on two different sets of clock target neurons and the efficient coupling between them enables Drosophila to keep proper phase relationships to light and temperature cycles (Miyasako, Umezaki and Tomioka, 2007) as found in nature.

L217…relationships to…

The modified sentence reads: Light and temperature, therefore, probably works on two different sets of clock target neurons and the efficient coupling between them enables Drosophila to keep proper phase relationships to light and temperature cycles (Miyasako, Umezaki and Tomioka, 2007) as found in nature.

L226 …(chordotonal organs),…

Corrected

L234…cycled at higher amplitudes…

L239 ...seem to…

Corrected

L275 …whereas it inhibits…

Corrected

L309-317 Please name first the abbreviations of the different splice forms of tim transcripts before explaining their respective functions.

Done

L320 …activity-rest rhythms…

Corrected

L332-338 please add all relevant citations

Added.

L340 …(Klose et al., 2021).

Corrected.

L345 …around dawn…

Corrected.

L359-363 Please discuss the results/discrepancies of/between findings of the Taghert lab that measured Ca++ levels in the somata and the finding by Klose et al., 2021.

We did not see any dramatic discrepancy between soma Ca++ and peptide release, in fact they are largely coherent, in our opinion.

L382…talk to

Corrected.

L386 ..insulin-like peptides (ILPs) and SIFamides…

Corrected.

L378 …Please discuss also the findings by Ruiz et al. 2021.

We thank the reviewer for pointing this out. Few sentences have been added to make the case for PI’s role in circadian output.

Reviewer 2 Report

Review written by Joydeep De and Abhishek Chatterjee represents a valuable overview of existing knowledge about perception of daily time in fruit flies. The authors briefly mentioned existence of the central and peripheral oscillators and then mostly focused on detailed explanation of circadian clock entrainment by light and temperature and the output of the clock.  For better orientation in the text, I would recommend the division into subchapters with headings describing their content. Although authors refer to almost 100 citations including the most recent ones, I miss there some important data. In the description of the temperature entrainment of the circadian clock I miss the information about genes nocte, pyrexia, IR25a, and about Trp channels, especially if last two are parts on the Figure 2. For details, see the recent review of Rebekah George and Ralf Stanewsky doi: 10.3389/fphys.2021.622545. Explanation of clock output pathway lack information about release of ion transport peptide (ITP).

The quality of the images does not reach the level of a scientific publication. Figure 1 is more like a child's drawing. Also, the schematic representation of the clock entry path may be more sophisticated, and all abbreviations must be explained in the figure legends. The separation of antenna in the different parts is unclear as well as the relation of antenna to other peripheral chordotonal organs.

In conclusion. After including the proposed adjustments and improving of Figures graphical design, the review will be suitable for publication in Insects journal.

Minor comments:

  • Add the italic font where it is necessary
  • There is no explanation for NLF-1, IP3, DH44+, DILP, and SIFa abbreviations

Author Response

Review written by Joydeep De and Abhishek Chatterjee represents a valuable overview of existing knowledge about perception of daily time in fruit flies. The authors briefly mentioned existence of the central and peripheral oscillators and then mostly focused on detailed explanation of circadian clock entrainment by light and temperature and the output of the clock.  For better orientation in the text, I would recommend the division into subchapters with headings describing their content.

Thanks for your valued feedback. We have now sectioned the longer “input” part into 3 subchapters.

Although authors refer to almost 100 citations including the most recent ones, I miss there some important data. In the description of the temperature entrainment of the circadian clock I miss the information about genes nocte, pyrexia, IR25a, and about Trpchannels, especially if last two are parts on the Figure 2. For details, see the recent review of Rebekah George and Ralf Stanewsky doi: 10.3389/fphys.2021.622545.

The information and the reference have been added.

Explanation of clock output pathway lack information about release of ion transport peptide (ITP).

The work on ITP, in our opinion is isolated (no mechanistic follow-ups), not conclusive, nor provide any significant explanation of the LD rhythms. We already have over 100 references; we think inclusion of the ITP-related work will not be sufficiently enriching to counterbalance the possibility of data overload.

The quality of the images does not reach the level of a scientific publication. Figure 1 is more like a child's drawing.

We drew it like a cartoon, as the purpose is to attract the attention of the reader and espouse the fact that circadian clocks are present all over the fly body.

Also, the schematic representation of the clock entry path may be more sophisticated, and all abbreviations must be explained in the figure legends.

Done.

The separation of antenna in the different parts is unclear as well as the relation of antenna to other peripheral chordotonal organs.

Done.

In conclusion. After including the proposed adjustments and improving of Figures graphical design, the review will be suitable for publication in Insects journal.

Thank you.

Minor comments:

  • Add the italic font where it is necessary: Done
  • There is no explanation for NLF-1, IP3, DH44+, DILP, and SIFa abbreviations: Done

Reviewer 3 Report

This is a very nice review on a hotly debated topic. How do animals actively perceive the time, anticipate periodic changes and, in the best case, gain an advantage or prevent disadvantages? The article is very well written and will be of great interest. 

Author Response

This is a very nice review on a hotly debated topic. How do animals actively perceive the time, anticipate periodic changes and, in the best case, gain an advantage or prevent disadvantages? The article is very well written and will be of great interest.

Thank you so much!